# The Impact of Modifying Sunitinib Treatment Scheduling on Renal Cancer Tumor Biology and Resistance

**DOI:** 10.3390/jcm11020369

**Published:** 2022-01-13

**Authors:** Harrison Sicheng Lin, Qiang Ding, Zsuzsanna Lichner, Sung Sun Kim, Rola Saleeb, Mina Farag, Ashley Di Meo, Pamela Plant, Mirit Kaldas, Georg Arnold Bjarnason, George Makram Yousef

**Affiliations:** 1The Keenan Research Centre for Biomedical Science, St. Michael’s Hospital, Toronto, ON M5B 1W8, Canada; slin345@uwo.ca (H.S.L.); colleen.ding@unityhealth.to (Q.D.); lichner_zsuzsi@yahoo.com (Z.L.); kimsspathology@gmail.com (S.S.K.); rola.saleeb@unityhealth.to (R.S.); dr.minafarag90@gmail.com (M.F.); ashley.dimeo@mail.utoronto.ca (A.D.M.); pamela.plant@unityhealth.to (P.P.); m_sawiris@hotmail.com (M.K.); 2Department of Pathology, Chonnam National University Medical School, Gwangju 61469, Korea; 3Department of Pathology, Unity Health Toronto, Toronto, ON M5B 1W8, Canada; 4Department of Laboratory Medicine and Pathobiology, University of Toronto, Toronto, ON M5S 1A1, Canada; 5Sunnybrook Odette Cancer Centre, Toronto, ON M4N 3M5, Canada; Georg.Bjarnason@sunnybrook.ca; 6Department of Paediatric Laboratory Medicine, The Hospital for Sick Children, Toronto, ON M5G 1X8, Canada

**Keywords:** renal cell carcinoma, sunitinib, resistance, treatment scheduling

## Abstract

With sunitinib treatment of metastatic renal cell carcinoma, most patients end up developing resistance over time. Recent clinical trials have shown that individualizing treatment protocols could delay resistance and result in better outcomes. We developed an in vivo xenograft tumor model and compared tumor growth rate, morphological, and transcriptomic differences between alternative and traditional treatment schedules. Our results show that the alternative treatment regime could delay/postpone cancer progression. Additionally, we identified distinct morphological changes in the tumor with alternative and traditional treatments, likely due to the significantly dysregulated signaling pathways between the protocols. Further investigation of the signaling pathways underlying these morphological changes may lead potential therapeutic targets to be used in a combined treatment with sunitinib, which offers promise in postponing/reversing the resistance of sunitinib.

## 1. Introduction

Renal cell carcinoma (RCC) is an epithelial malignancy of the renal tubules [1]. It affects 2–3% of the world population with the highest incidence rate in Europe (7–34/100,000 males and 3–15/100,000 females) [1]. The incidence of RCC is increasing at a rate of 2–3% every year worldwide, making it one of the 10 most common cancers in North America [2]. Clear cell renal cell carcinoma (ccRCC) accounts for 85% of adult RCC [3,4]. Loss of function of *VHL* at 3p25 is observed in most cases [5]. The pathogenesis of ccRCC is deeply related to the loss of function of *VHL*, which results in a pro-angiogenic gene expression signature by the destabilization of hypoxia-inducible factors [5]. Microscopically, ccRCC is characterized by an extensive network of thin-walled, staghorn-shaped vasculature [5]. Another important pathway implemented in ccRCC pathogenesis is the PI3K/AKT/MTOR pathway, which is currently targeted in therapies [5]. Mutation in *SETD2* and SWI/SNF chromatin remodeling complex is also characterized in ccRCC [5].

Radical or partial nephrectomy is the treatment for primary ccRCC. For metastatic ccRCC, oral vascular endothelial growth factor receptor TKIs, including “sunitinib”, have significantly improved outcomes for patients [6]. TKI monotherapy has now been replaced in the first line setting by combination immunotherapy [7], or immunotherapy given in combination with TKIs [8,9,10]. TKIs continue to play a major role in later lines of therapy. Sunitinib works to inhibit the phosphorylation of a number of receptor tyrosine kinases, such as vascular endothelial growth factor receptors 1, 2 and 3, platelet-derived growth factor receptors α and β, stem cell factor receptor, Fms-like tyrosine kinase-3, the glial cell-line derived neurotrophic factor receptor, and the colony stimulating factor receptor. Consistent with its multi-targeted profile, sunitinib works to promote tumor regression and inhibit tumor growth, deter pathologic angiogenesis, and prevent metastatic progression of cancer [11].

The prolonged use of sunitinib leads to clinical morbidities such as hypertension, oral mucositis, hand-foot syndrome, diarrhea, hematological toxicity, and fatigue [12]. Additionally, most patients develop resistance to sunitinib and eventually succumb to the disease; the median progression-free survival under a sunitinib treatment ranges 3–14 months in the different risk groups [10]. The causes for sunitinib resistance are yet to be fully elucidated yet, work from our lab and others has shown that resistance is a late manifestation of early, treatment-induced histo-molecular alterations [13]. Our previous findings also suggest that sunitinib-resistant RCC cells can be histologically identifiable while the tumor is still treatment-sensitive [3].

A recent single arm phase-II clinical trial has shown that individualized sunitinib therapy may be more effective and safer compared to the traditional treatment regime [12]. In this trial, toxicity was used as a surrogate for adequate drug exposure. Sunitinib dose and number of days on therapy were individualized based on toxicity aiming for ≤grade-2 toxicity (oral mucositis, diarrhea, hand-foot syndrome, neutropenia, thrombocytopenia, and fatigue) with dose escalation in patients with minimal toxicity. Another trial showed that metastatic RCC patients who were switched to a modified 2/1 schedule of sunitinib showed better safety profile compared with that seen with the initial 4/2 schedule and concluded that alternative schedules, such as 2-week-on treatment and 1-week-off (2/1 schedule), might improve tolerability [14].

In the present study, we hypothesize that varying treatment schedule and dosage of sunitinib can delay tumor resistance to the drug and prolong progression-free survival compared to the traditional schedule. We gave sunitinib at different dosages and administration schedules in vivo using a xenograft mouse model and examined the morphologic and molecular differences of the resulting tumors between the treatment groups as a tool to examine the effects of altering the schedule of treatment on the development of resistance.

## 2. Materials and Methods

### 2.1. Cell Culture and Mouse Models

All animal studies were performed in compliance with the Animal Care Committee of St Michael’s Hospital and the Canadian Council of Animal Care. The 786-0 kidney cancer cell line was obtained from American Type Culture Collection (Manassas, VA, USA) and was cultured per the distributor’s description. An amount of 5 × 10^5^ cells were injected into NOD/SCID (NSG) mice (6–8 weeks old, female; The Jackson Laboratory, Bar Harbor, ME, USA) subcutaneously. A 1:1 ratio of Matrigel (Thermo Fisher Scientific, Waltham, MA, USA) and single-cell suspension were contained in the injection [3]. Mice were treated with varied dose of sunitinib or citrate buffer (vehicle-treated group) by gavage every day, as detailed below [3]. A total of 51 animals were used in the study: 3 controls and 16 animals of each of the three treatment groups. Some animals died throughout the follow up time. 

### 2.2. Sunitinib Administration Protocols

Sunitinib was administered by gavage. Mice were randomized to the following four groups: (1) Control: no treatment, (2) Traditional: Sunitinib for 4 weeks of continuous treatment and 2 weeks of break (comparable to standard clinical regimen [15] (3) Alternative protocol 1: Sunitinib 50 mg/day, 2 weeks of continuous treatment and 1 week of break and (4) Alternative protocol 2 (high dose): 75 mg Sunitinib/day, 2 weeks of continuous treatment and 1 week of break. Animals were observed for a total of 12 weeks. Every experimental group started with 11–13 mice.

### 2.3. Tumor Growth Assessment

Tumor size was measured once the tumors become palpable and monitored by manual caliper daily [3]. Tumor volume was estimated by the following equation [3]:$$\mathrm{Volume}={\left(\mathrm{width}\right)}^{2}\times \frac{\mathrm{length}}{2}$$

Tumor volume was estimated by using the equation above and the average volume per week was calculated for every group.

### 2.4. Histological Assessment

Three mice from every experimental group were sacrificed for histological examination. Tumors and organs (liver, kidneys, lungs, spleen, stomach, and any area suspicious for metastasis) were collected or fixed in 10% formalin. Formalin-fixed paraffin-embedded tissues were sliced at 4–6 μm sections and stained with hematoxylin and eosin. The stained sections were assessed by two pathologists (RS and SK) [3]. Slides were scanned by an Aperio scanner (Leica Biosystems, Buffalo Grove, IL, USA) for image quantification analysis.

### 2.5. RNA Isolation and Quantification

Three mice from every experimental group were sacrificed for mRNA expression analysis. Tumors were snap-frozen at −80 °C for RNA isolation. Total RNA was isolated using Qiagen RNeasy Mini Kit (#74104, Qiagen, Germantown, MD, USA), following the manufacturer’s recommendations. RNA quantity and integrity was assessed using the RNA 6000 Nano Assay and Agilent 2100 Bioanalyzer (Agilent Technologies Inc., Santa Clara, CA, USA). Samples with RIN greater than 7 were used for RNA-Seq analysis. mRNA sequencing libraries were constructed as per the recommendations of the TruSeq mRNA protocol (Illumina, San Diego, CA, USA).

### 2.6. Next Generation Sequencing Analysis of mRNA Transcripts

RNA-Seq was performed on snap-frozen tumors from traditional and alternative treatment groups. First strand cDNA was synthesized using 1 μg of total RNA, using the Illumina TruSeq mRNA library prep kit, following the manufacturer’s recommendation. Libraries were sequenced on the Illumina NextSeq 550 system (Illumina, San Diego, CA, USA). The targeted read counts were 20–35 million total reads per sample. Raw FastQ reads files were assessed and adapter trimming processed using the RNA-Seq Alignment app (Basespace, Illumina), and reads with Phred scores > 30 were retained. The resultant quality-trimmed reads were aligned to the hg38 (GrCH38.83) build of the human genome using the STAR aligner app (BaseSpace, Illumina). Transcript abundance quantification was performed using Cufflinks Assembly & DE analysis apps (BaseSpace, Illumina).

### 2.7. Statistical and Bioinformatics Analysis

The differences in tumor sizes between groups were tested using the t-test (two sample assuming unequal variance). The compared groups were control against traditional, alternative 1 against traditional, alternative 2 against traditional and alternative 1 against alternative 2. Gene expression profiles were evaluated by GenePattern RNA Seq modules and GSEA (Gene Set Enrichment Assay) (https://www.genepattern.org/ (accessed on 20 August 2021), https://www.gsea-msigdb.org/gsea/index.jsp (accessed on 20 August 2021)).

To gain a better understanding of the molecular events that led to aggressive behavior and sarcomatoid dedifferentiation, genes with differential falling outside of the inflection point were analyzed using the GSEA. Enrichment gene sets results were visualized by Cytoscape (https://cytoscape.org/, accessed on 21 August 2021). We selected late treatment cases to evaluate the molecular and gene pathways involved in the changes [16,17].

## 3. Results

### 3.1. Alternative Treatment Scheduling Delayed Tumor Growth and Drug Resistance

The mean growth curves of all four groups are summarized in Figure 1. Tumor volumes were followed for 12 weeks. As expected, sunitinib treatment resulted in tumor stability/slower growth rate in all treatment groups. The traditional treatment showed significant difference (*p* < 0.05) when compared with no treatment.

The control group showed the highest growth rate, as expected. Tumors started getting palpable at the second week. The traditional scheduling group (50 mg/day for 4 weeks of continuous treatment and 2 weeks of break) showed resistance around the 6th week and rapid tumor growth by the 9th week. The two alternative scheduling groups (50 mg/day (Alt 1) or 75 mg/day (Alt 2) for 2 weeks of continuous treatment and 1 week of break) showed slower tumor growth rates and did not show signs of resistance till the end of the experiment. There was no significant difference when different alternative schedule dosages were compared (i.e., 50 mg vs. 75 mg). Each graph shows the average of three animals.

The slope of traditional treatment (m = 527.61) showed stable disease with minimal tumor growth until about the 6th week. Rapid growth of the tumors occurred at approximately the 9th week and was fully manifested at the 10th week, indicating sunitinib resistance.

We observed a significant slower tumor growth rate between traditional and alternative scheduling. Compared with the control group (no treatment), and the traditional group, the slopes of the two-alternative scheduling treatments were nearly half (m_1_ = 218.46 and m_2_ = 273.46), which indicates the alternative treatment significantly slowed down tumor growth rate, and furthermore, there were no signs of resistance until the end of the experiment (12 weeks). There was, however, no significant difference in tumor growth rate between the two alternative treatment groups (*p* > 0.05 throughout 12 weeks of the experiment) (Figure 2).

Tumors were palpable at the second week and at 6, 9 and 11 weeks. The traditional group was treated with 50 mg sunitinib/day for 4 weeks of continuous treatment and 2 weeks of break). The alternative groups were treated by either 50 mg/day (alt1) or 75 mg/day (alt 2) for 2 weeks of continuous treatment and 1 week of break. Tumor volumes were significantly larger in the traditional group. We observed no significant difference of tumor growth when comparing different dosages in the alternative group.

### 3.2. The Effect of Alternative Treatment Scheduling on Tumor Morphology and Behaviour

We compared the histomorphology of tumor xenografts subjected to the traditional treatment (Figure 3) and alternative treatment (Figure 4) at different time intervals. The traditional treatment showed early changes of aggressive behavior (Figure 3A) including greater extent of vasculature and spheroid formation, bizarre large nuclei and multi-nucleated cells (which indicates enrichment of stem-cell and EMT properties) (Figure 3B), irregular borders infiltrating into the surrounding tissue (Figure 3C), in agreement with our previous results showing early changes induced by sunitinib [18]. Mice from this group showed also remote metastatic lesions in lung (Figure 3D,E) and liver (Figure 3F,G).

Histological examination of the alternative treatment scheduling cohort was evaluated after two and three cycles of treatment (Figure 4A). A cycle was defined as three weeks of treatment which either represents continuous treatment for the traditional group or two weeks of treatment and one week of no treatment for the alternative groups. After two cycles (six weeks) and three cycles (nine weeks), tumor sections showed significantly larger islands of necrosis (Figure 4B) (indication of treatment effect) compared to traditional scheduling (*p* < 0.05), significantly less vascularization (*p* < 0.05), and very minimal component of spindling (a marker of sarcomatoid changes).

Additionally, alternative treatment groups showed a significantly higher proportion of cells with preserved RCC morphology (Figure 4C) (rounded to oval cells with cytoplasmic clearing) whereas the traditional group demonstrated aggressive behavior indicated by multinucleated giant cells, bizarre nuclei and spindle shaped formation (sarcomatoid changes).

In terms of aggressive tumor behavior, lung and liver metastasis were observed in both the traditional and the alternative treatment groups, but tumors with traditional treatment started developing metastasis at a much earlier date (at 6 weeks) and showed significantly higher number and larger size of metastatic deposits. The interrupted treatment protocol resulted in a smaller number of metastatic deposits in the liver and the lung compared to continuous treatment or the control groups after 12 weeks (*p* < 0.05). When comparing the different dosage of the alternative treatments, the higher dosage (75 mg) clearly produced more confluent areas of necrosis, indicating more effective treatment than the regular classic dosage (50 mg), even at 12 weeks, although, ultimately, this did not translate into a significantly smaller tumor size (as discussed above).

Comparing animal health during treatments, there were significantly fewer side effects as manifested by less hair depigmentation, depression (less motion), etc. in the alternative groups as compared to the traditional treatment. It should also be noted that the mice in the alternative group were able to tolerate a higher dosage of the treatment with fewer side effects. Together, these results indicate that the drug “breaks” (in the alternative treatment) resulted in greater efficiency, with fewer side effects, and enabled the animal to tolerate higher doses, and significantly delayed the development of aggressive features and the development of resistance.

### 3.3. Transcriptomic Profile and Behavior Were Different between Traditional and Alternative Treatment Groups

Following sequencing, FastQ and sequencing QC were evaluated and samples with a sequencing depth greater than 30 were selected for comparison. There was a significant expression profile difference between the traditional and alternative groups, as shown in Figure 5.

GSEA was performed to compare the expression pattern of the alternative vs. traditional scheduling groups using C2 (curated gene sets), the H (hallmark gene sets) collections in MSigDB (Molecular Signature Database). We interrogated the C2 collection to find related pathways, and the H collection (more accurate in reducing noise and redundancy ) for identifying the significantly altered biological processes, as previously completed [19]. We found 48/50 gene sets are upregulated in the traditional group compared to the alternative scheduling; of these, 13 gene sets were significant with a False Discovery Rate (FDR) < 25%, 8 gene sets are significantly enriched at nominal *p* < 0.01, and 12 gene sets were significantly enriched at nominal *p* < 0.05. The eight highly enriched (significant) pathways were protein secretion, androgen secretion, heme metabolism, mitotic spindle, oxidative phosphorylation, mTORC1 signaling, early estrogen response, Myc targets, bile acid metabolism, UV response DN, G2M checkpoint, and PI3K-AKT-mTOR signaling (Table 1). Many of these, like mTOR (Figure 5B) and cMyc are reported to contribute to tumor progression and aggressive behavior, as discussed below. The relative gene numbers involved in each of these pathways is illustrated in Figure 5C. Signaling enrichment results were reproducible using an alternative/additional Reactome database analysis that showed that these pathways form functional interaction networks (Figure 5D). There were two gene sets that were upregulated in the alternative treatment group compared to the traditional treatment, but not significantly enriched. These were myogenesis and epithelial mesenchymal transition pathways (Data not shown).

## 4. Discussion

Sunitinib and other TKIs remain important treatments for metastatic renal cell carcinoma. The main limitation of its usage is the high prevalence of developing resistance over the time course of treatment. The mechanisms of resistance that develop to TKIs such as sunitinib are poorly understood. A recent clinical trial by Bjarnason et al. showed that an “individualized” dosing of sunitinib might improve outcomes for patients [12]. The purpose of our current study is to examine if altering treatment scheduling will delay the development of resistance in vivo, and to dissect the histo-morphological and molecular attributes that might underlie this phenomenon.

Our results are consistent with the Bjarnason et al. clinical trial, where shorter sunitinib exposure followed by a “break” in treatment, results in improved survival [12]. Altering the duration of sunitinib treatment significantly reduced the rate of tumor growth (*p* < 0.05) throughout the course of the study. It also altered tumor morphology, and the molecular signatures underlying tumor behavior.

We have previously reported that sunitinib induces early histo-molecular changes in renal cancer cells that can contribute to resistance [3]. Other studies have shown that the processes underlying (reversible) epithelial to mesenchymal transition (EMT) may be associated with acquired tumor resistance to TKIs in patients with ccRCC [20]. Taken together, it is possible that an interrupted treatment protocol can help reverse, or at least delay, some of the changes that lead to resistance. In the current study, delayed aggressive behavior (as characterized by multinucleated giant cells, bizarre nuclei and spindle shaped formation) was observed in the alternative treatment along with extensive necrosis of the tumor cells, less vasculature and irregular well-defined borders. These are all signs of better treatment efficiency, while on the other hand, the traditional treatment had shown early sign of aggressive behavior, more vasculature and irregular borders infiltrated the surrounding tissue spheroid formation.

There are currently no clinical biomarkers to predict sunitinib response. Most respondents develop resistance through reversible mechanisms that are poorly understood. We previously identified miRNAs that can predict sunitinib response [21] and showed miRNA involvement in development of resistance [22]. In the current study, we identified a number of pathways that can contribute to molecular changes associated with resistance and that represent potential drug targets. Further testing is required to examine the utility of key molecules in these pathways to monitor and predict development of resistance.

Sunitinib has significant impact on tumor cell programming in addition to its anti-angiogenic role on endothelial cells. In our differential gene expression analysis and GSEA, most enriched genes are involved in tumor aggressiveness and drug resistance including ABHD2, ABCC4, CLN5 intracellular trafficking protein, and insulin like growth factor 2 receptor. ABHD2, an androgen target gene, was reported to promote prostate cancer cell proliferation and migration [23]. ABCC4 is associated with resistance to drugs in solid tumors and was shown to contribute to the aggressiveness of Myc-associated epithelial ovarian cancer [24].

Furthermore, the traditional treatment group showed increased expression of genes involved in pathways that are highly associated with tumor aggression. Remarkably, some cancer stem cell genes were also found in this group, such as PROM1 (also known as CD133) which is a member of a prominent family of pentaspan transmembrane glycoproteins of murine neuroepithelial origin (typically located in plasma membrane protrusions) [25]. CD133 is found in embryonic stem cells, normal tissue stem cells, stem cell niches, circulating endothelial progenitors as well as cancer stem cells [26,27].

A previous study described an upregulation of lipid biosynthesis in the sunitinib-resistant 786-O kidney cancer cell line and was suggested to accelerate membrane construction in both enlarged nuclei and lysosomes [28]. Increased expression of CADM1 resulted in significant inhibition of motility and invasiveness of melanoma cells [29].

Our findings are consistent with reports of several other genes that are implicated in progression of cancer. For example, Epithelial Membrane Protein 1 (*EMP1*) is expressed in high levels in human cancers and was shown in vitro to reduce cell migration and invasion, and was also shown to increase apoptosis and caspase-9 expression in carcinoma of the nasopharynx, stomach, breast and prostate [30]. Other highly enriched genes in the traditional regimen group included SAMD, which has been shown to be an unfavorable prognostic marker in kidney cancer [31], and CYP8B, a cholesterol metabolizing enzyme reported as being an unfavorable prognostic marker in colorectal cancer [32]. Thus, our mouse xenograft model of RCC and the application of the traditional sunitinib regime renders a canonical profile of genes associated with poor outcomes in various cancers.

In the alternative treatment groups, two gene sets upregulated; however, the differences were not significant compared to the traditional treatment. These two sets are pathways associated with myogenesis and EMT.

The current study shows that an alternative treatment scheduling may delay resistance; however, it did not significantly reduce the chance of metastasis. A further study on how to inhibit EMT is warranted.

## Figures and Tables

**Figure 1 jcm-11-00369-f001:**
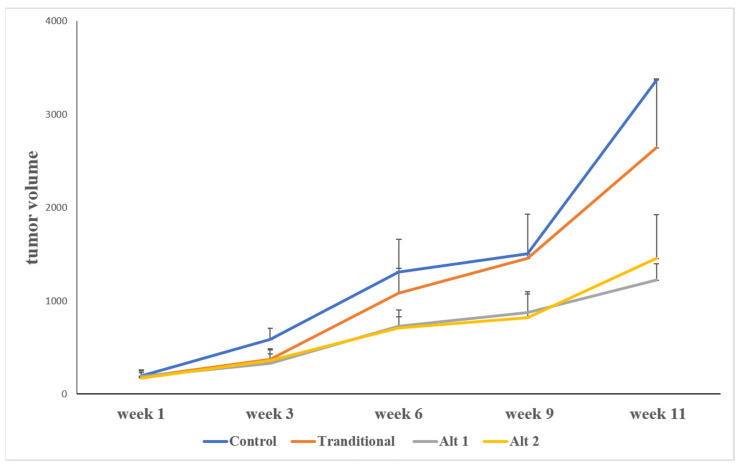
Comparison of tumor growth rates between the different scheduling groups.

**Figure 2 jcm-11-00369-f002:**
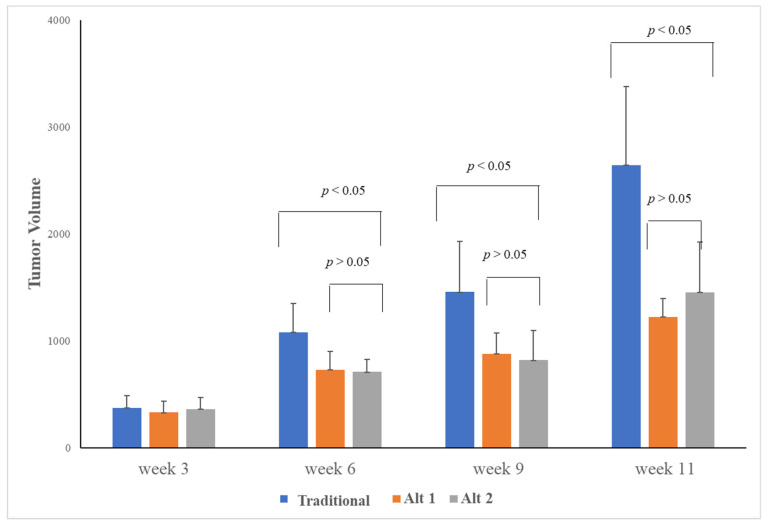
Bar graph showing tumor volumes at different time points.

**Figure 3 jcm-11-00369-f003:**
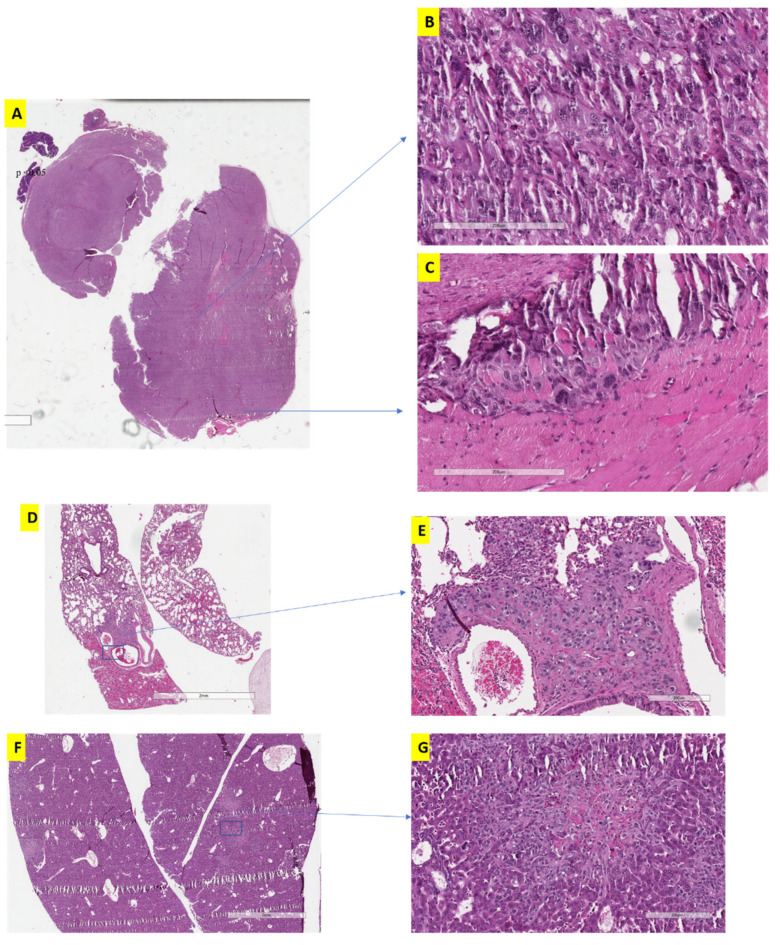
Histopathologic changes in the traditional scheduling regimen after three cycles of treatment (9 weeks). (**A**) lower power magnification showing infiltrative tumor border into surrounding tissues; (**B**) high power magnification showing giant cells with bizarre nuclei and multi-nucleation. (**C**) high power magnification showing infiltrative border with tumor cells invading into the surrounding tissues. (**D**) remote metastatic lesion in lung. (**E**) higher magnification of (**D**) showing metastatic deposits in lung. (**F**,**G**) metastatic deposits in the liver (low and high magnification, respectively).

**Figure 4 jcm-11-00369-f004:**
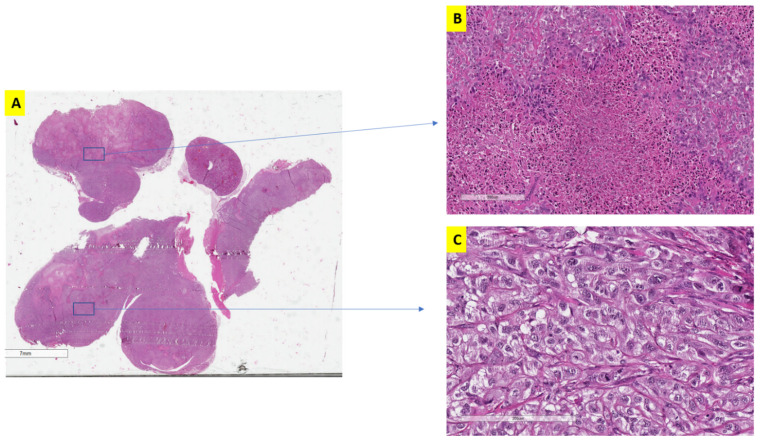
Histomorphology of Alternative treatment xerographs. Alternative treatment xerographs showed significantly larger islands of necrosis (**A**,**B**) and more preservation of renal tumor cell morphology (**C**).

**Figure 5 jcm-11-00369-f005:**
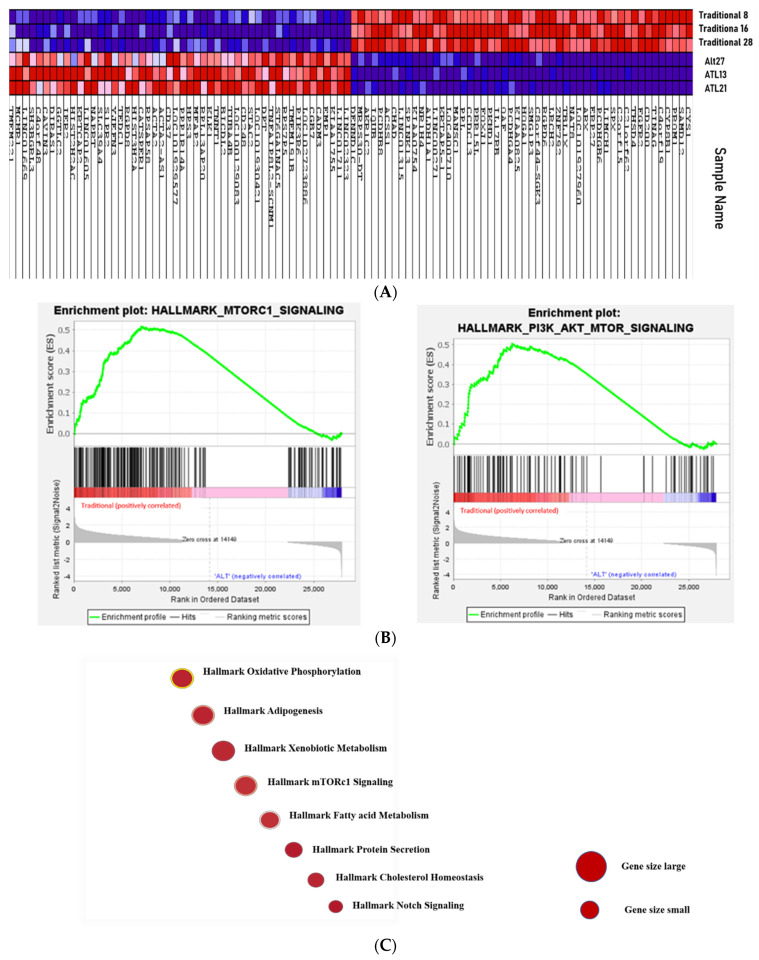
(**A**) Transcriptome profile comparison between the traditional and the alternative groups. The expression of a number of genes was significantly different between the two groups. Red = upregulated genes. Blue = downregulated genes; (**B**) mTOR signaling and PI3K/AKT pathways were significantly enriched in traditional; (**C**) significant enriched the signaling pathways in traditional treatment. The size of the circle denotes the relative number of genes in each pathway; (**D**) signaling enrichment analysis performed using Reactome analysis showed comparable pathways to the H collection analysis. It showed additionally that groups of these pathways form Reactome functional interaction networks, as indicated by the connecting lines.

**Table 1 jcm-11-00369-t001:** Significantly enriched gene sets in the traditional treatment group, compared to alternative scheduling protocols.

Gene Set Name	Size	Enrichment Score	Normalized Enrichment Score	NOM *p*-val	FDR *q*-val
HALLMARK_PROTEIN_SECRETION	96	0.644088	1.524993	0.0001	0.00576
HALLMARK_ANDROGEN_RESPONSE	99	0.622935	1.48529	0.0002	0.00625
HALLMARK_HEME_METABOLISM	195	0.54383	1.322556	0.0005	0.1018
HALLMARK_MITOTIC_SPINDLE	198	0.510461	1.245078	0.001	0.171488
HALLMARK_OXIDATIVE_PHOSPHORYLATION	200	0.513705	1.252967	0.002	0.212878
HALLMARK_MTORC1_SIGNALING	197	0.516115	1.25293	0.002	0.177398
HALLMARK_ESTROGEN_RESPONSE_EARLY	198	0.513156	1.244218	0.006	0.15162
HALLMARK_MYC_TARGETS_V1	196	0.499329	1.21534	0.01	0.171292
HALLMARK_BILE_ACID_METABOLISM	112	0.516231	1.230885	0.019076	0.162642
HALLMARK_UV_RESPONSE_DN	142	0.506569	1.219168	0.023	0.17801
HALLMARK_G2M_CHECKPOINT	196	0.480435	1.162778	0.034	0.296556
_PI3K_AKT_MTOR_SIGNALING	104	0.50546	1.209761	0.04911	0.160687

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
