# Peer review of "The Impact of Modifying Sunitinib Treatment Scheduling on Renal Cancer Tumor Biology and Resistance"

_jcm, 2022, doi:10.3390/jcm11020369_

Round 1

Reviewer 1 Report

The manuscript titled ‘The Impact of Modifying Sunitinib Treatment Scheduling on Renal Cancer Tumor Biology and Resistance’ submitted by Lin et.al., investigated the Sunitinib’s alternative treatment method for renal cancer in vivo xenograft tumor model. Authors have shown that this alternative treatment method could delay cancer progression compared to traditional treatment strategy. Authors have observed a significant slower tumor growth rate between traditional and alternative treatment method. The traditional method showed development of resistance after 6th week and a rapid tumor growth by the 9th week. However, no such resistance and increase in growth rate was observed in suggested alternative method.

Overall the study is interesting and the manuscript is well written without exaggerating the findings. As authors mentioned more studies are needed to reach any fruitful conclusion, thus, it is required this research should be available for the research community for further study.

Author Response

We would like to thank the reviewer for his/her overall very positive comments about the manuscript.  The reviewer praises the value of the manuscript and the importance of the results.

No criticism was raised by the reviewer. 

Reviewer 2 Report

The Authors have investigated a relevant topic within renal cell carcinoma biology with potential implications as treatment management in clinical practice. 

I suggest the following modifications that may improve the paper:

page 1, line 38: please correct "pathogenies"

page 2, line 91: please specify sunitinib dosage a day

page 3, line 135: please correct "GESA" in GSEA

page 4:

-line 149: please introduce into the legend the description for each group, and correct "tranditional"

-line 150: please remove "Figure 1"

-line 155: please modify "with" and delete the round brackets

page 5, line 170: please report into the legend the description for each group. Line 171: please remove "Figure 2"

page 6, lines 186-187: please remove the text (repetition?). Line 196: please rephrase the sentence without round brackets and put attention for three (?) weeks

page 7, line 209-210: please join the two sentences. Line 219: please correct "amore". Lines 227-230: this period is misplaced, please put it into Discussion section

page 8, line 238: please control the legend of the Figure 5A. This experiment should be explained in the text.

page 12, line 315: please control the punctuation before "Remarkably". Line 337: please remove (epithelial mesenchymal transition)

Author Response

the reviewer raised a number of  important corrections that need to be fixed. we addressed them all as follows:

page 1, line 38: please correct "pathogenies"

the typographical error is now fixed, as advised 

page 2, line 91: please specify sunitinib dosage a day

we have now specified the dosage as 50 or 75 mg/day, as instructed by the reviewer

page 3, line 135: please correct "GESA" in GSEA

the typographical error is now fixed, as advised 

page 4:

-line 149: please introduce into the legend the description for each group, and correct "tranditional"

the description of each group is now introduced and the typographical mistake is fixed. 

-line 150: please remove "Figure 1"

Figure 1 is now removed 

-line 155: please modify "with" and delete the round brackets

done, as instructed by the reviewer

page 5, line 170: please report into the legend the description for each group.

done

Line 171: please remove "Figure 2"

done as instructed

page 6, lines 186-187: please remove the text (repetition?).

done

Line 196: please rephrase the sentence without round brackets and put attention for three (?) weeks

page 7, line 209-210: please join the two sentences. Line 219: please correct "amore". Lines 227-230: this period is misplaced, please put it into Discussion section

done

page 8, line 238: please control the legend of the Figure 5A. This experiment should be explained in the text.

done as advised 

page 12, line 315: please control the punctuation before "Remarkably". Line 337: please remove (epithelial mesenchymal transition)

done

Round 2

Reviewer 2 Report

The Authors have improved the article which now lends itself to being reconsidered by the Journal for publication

Author Response

Line 118, please add the initials of the 2 pathologists after “were assessed by 2 pathologists”

done as instructed

Line 147 please add the link to the GSEA

Link added. Please note that the CTL indicates the traditional group and was used as a control for the purpose of this analysis.

Line 167: Please indicate which one is the Alt 1 and which one is the Alt 2

done as instructed

Line 170 Each graph shows the average of three animals whereas line 158 & 159 says Tumor volumes were calculated from a total of 51 mice for 12 weeks. Please clarify the difference in numbers and please add this clarification to the manuscript.

we have now moved this to the materials and methods section to avoid confusion. We indicated that there were 3 controls and 4 from each group.

Line 180 Please put “Figure 2. Bar graph showing tumour volumes at different time points” below or above its corresponding figure.

Done

Line 243 to 246 Please, clarify if these lines were moved to the discussion and should be deleted here.

We would like to leave it as a summary of the results. We leave the final decision to the editor

Figure 5A: Please add the colour code for figure 5A, (i.e. what is indicated by the blue and red colours?). Also, please, confirm if the label of the sample name is correct here or should be changed to the gene name?

colour coding is now indicated. the labelling is correct as it is.
